

# Vulnerability curves versus vulnerability indicators: application of an indicator-based methodology for debris-flow hazards

Maria Papathoma-Köhle[1]

[1]Institute for Mountain Risk Engineering, University of Natural Resources and Life Sciences, Vienna, 1190, Austria

*Correspondence to*: M. Papathoma-Köhle (maria.papathoma-koehle@boku.ac.at)

**Abstract.** The assessment of the physical vulnerability of elements at risk as part of the risk analysis is a very important aspect for the development of strategies and structural measures for risk reduction. Understanding, analysing and quantifying, if possible, physical vulnerability is a prerequisite for designing strategies and adopting tools for its reduction. The most common

methods for assessing physical vulnerability are vulnerability matrices, vulnerability curves and vulnerability indicators, however, in most of the cases these methods are used in a conflicting way rather than in combination. The article focuses on two of these methods: the vulnerability curves and the vulnerability indicators. Vulnerability curves express physical vulnerability as a function of the intensity of the process and the degree of loss. However, a considerable amount of studies argue that vulnerability assessment should focus on the identification of these variables that influence the vulnerability of an

element at risk (vulnerability indicators). In this study, an indicator-based vulnerability methodology for mountain hazards including debris flow (2012) is applied in a case study for debris flows in South Tyrol where in the past a vulnerability curve has been developed. The relatively "new" indicator-based method is being scrutinised and recommendations for its improvement are outlined. The comparison of the two methodological approaches and their results highlight their weaknesses and strengths, show clearly that both methodologies are necessary for the assessment of physical vulnerability and emphasise

the need for a "holistic methodological framework" for physical vulnerability assessment.

## 1 Introduction

Climate and environmental change are expected to alter the patterns of risk in mountain areas. On one hand, the frequency, magnitude and spatial extend of natural hazards is expected to change, on the other hand, extensive development and changes in land use and land cover will certainly alter the spatial pattern of the vulnerability of the elements at risk. Especially in the

Alps, the influence of climate change on geomorphological hazards as well as their monitoring and modelling is a major issue (Keiler et al., 2010). It is clear that although predicting, monitoring and assessing the hazardous process is essential, the analysis of the vulnerability of the elements at risk may be the key to risk reduction. To address vulnerability in a holistic way, all its dimensions (social, economic, physical, environmental, institutional) should be addressed and analysed. However, herein the focus is solely on the physical vulnerability of buildings. Physical vulnerability is often considered to be the degree of loss





following a disastrous event. Nevertheless, the characteristics of the elements at risk that constitutes them susceptible to harm are often overlooked and have to be further investigated. The most common method for assessing vulnerability is the development of vulnerability curves that ignore the characteristics of the buildings since they focus only on the intensity of the process and the corresponding loss. Nevertheless, in the present study an indicator methodology is applied in Martell (S.Tyrol,

Italy) for debris flows. The same case study area has been used in the past for the development of a vulnerability curve based on damage data from a debris flow event in 1987 (Papathoma-Köhle et al., 2012). By comparing the results of the two methods, the advantages and disadvantages of the indicator based method can be highlighted and recommendations for its further development and improvement can be made.

## 2. Physical vulnerability assessment: the PTVA method and the use of indicators

Due to the multi-dimensional nature of vulnerability a long list of definitions can be found in the literature varying from general ones to more dimension specific ones. As far as the physical vulnerability is concerned, the definition of UNDRO (1984) ("vulnerability is the degree of loss to a given element, or set of elements, within the area affected by a hazard. It is expressed on a scale of 0 (no loss) to 1 (total loss)") is in conflict with other more general definitions presenting vulnerability as susceptibility to harm or a result of a combination of characteristics of the element at risk. For example, according to (UNISDR,

2009) vulnerability is defined as "the characteristics and circumstances of a community, system or asset that makes it susceptible to the damaging effects of a hazard". It is, therefore, clear that the lack of common definition for physical vulnerability results to the lack of universal methodology for its assessment.

### 2.1 Methods for assessing physical vulnerability

The variety of methodologies and concepts regarding vulnerability in general has been highlighted and demonstrated in several publications (e.g. Fuchs et al.(2012), Papathoma-Köhle *et al.*(2011)) . The most common approaches for assessing vulnerability in general are vulnerability matrices, vulnerability curves and indicator-based approaches (Kappes et al., 2012). Vulnerability matrices provide only qualitative information regarding vulnerability based on descriptions of damage patterns.

As far as physical vulnerability assessment is concerned, developing vulnerability curves is the most common method for assessing physical vulnerability as far as mountain hazards are concerned (Papathoma-Köhle et al., 2011). The vulnerability of buildings to natural hazards is determined by a number of attributes such as building material, size, condition etc. (Tarbotton et al., 2015). However, these attributes and their combination vary from building to building. This makes the data collection a time-consuming process which requires a detailed building-to-building investigation. Consequently, decision-makers and

practitioners, use data from past events to develop empirical vulnerability models such as vulnerability curves (Tarbotton *et al.*, 2015). Another reason that makes vulnerability curves so popular among practitioners is that they connect directly the intensity of a process with the corresponding degree of loss, providing concrete quantitative results and translating potential events into monetary damage. However, the reliability of the vulnerability curves is directly dependent to the quality and the



quantity of data that are used for their production. Tarbotton *et al.* (2015) reviewing methods for the development of vulnerability functions for tsunami, suggest that the accuracy and reliability of the results of empirical vulnerability curves depend on a series of factors including: the survey method for the data collection, the accuracy of the data regarding the building damage as well as, the building characteristics and the statistical method used for the analysis of the data. In more

detail, according to the same authors, the survey method may be remote (e.g. the building damage and type may be identified by the use of ortho-photos) or on site (field survey). The remote survey methods are faster, cheaper but inaccurate whereas the field survey is accurate but time-consuming and expensive. Moreover, for the development of the vulnerability curves the intensity has to be expressed in a measurable way (e.g. height of deposit or impact pressure). The choice of the parameter of the intensity that will be used for its expression influences the result (Tarbotton et al., 2015). Moreover, Tarbotton *et al.* (2015),

focusing on tsunamis, point out that the uncertainty of the results is increasing with the use of interpolation in order to identify the intensity of a process on individual buildings. Last but not least, the number of buildings used for the development of a vulnerability curve also influences the accuracy and validity of the results. In case of earthquakes or floods the assessment of the intensity per building is easier and the number of affected buildings is large. In these cases, it is relatively uncomplicated to develop reliable vulnerability curves. For other hazard types, such as rock falls, this is not the case since a single event

affects only a limited amount of elements at risk and the assessment of the process intensity on each of them is challenging. Moreover, vulnerability curves do not consider the different features of the buildings or characteristics related to their location and surroundings. Although they provide concrete information regarding the loss, they do not provide information concerning the drivers of vulnerability and eventually, potential ways of reducing it. Nevertheless, vulnerability curves have been used very often for the assessment of building vulnerability to debris flow (Fuchs *et al.* (2007), Quan Luna *et al.* (2011), Totschnig

*et al.* (2011), Papathoma-Köhle *et al.* (2012), Totschnig and Fuchs (2013)).

The lack of empirical data for the development of vulnerability curves often leads to the development of indices which are based on the selection, weighting and aggregation of vulnerability indicators. Indicator-based methodologies have been used mainly for other vulnerability dimensions such as social or economic vulnerability. However, recently, a considerable amount

of studies is available that make use of vulnerability indicators for the assessment of other vulnerability dimensions (e.g. physical). Vulnerability indicators according to Birkmann (2006) are "variables which are operation representations of a characteristic quality of the system able to provide information regarding the susceptibility, coping capacity and resilience of a system to an impact of an albeit ill-defined event linked to a hazard of a natural origin". The importance of developing and using indicators was also stressed in Hyogo Framework by identifying as a key activity the development of "systems of

indicators of disaster risk and vulnerability at national and sub-national scales that will enable decision-makers to assess the impact of disasters on social, economic and environmental conditions and disseminate the results to decision-makers, the public and populations at risk" (UN, 2007). However, using indicators to assess vulnerability (any dimension of it) may be problematic due to a number of reasons (Barnett et al., 2008). In more detail, Barnett et al. (2008) suggests that the use of vulnerability indicators and indices for national scale is less meaningful, whereas for larger scale it might lead to policy relevant



results, however, it still bears many uncertainties. The same authors also stress that challenges are related to the selection of indicators, their standardisation, the availability of required data, their weighting and the methods of aggregation for the development of a vulnerability index, concluding that empirical investigation (e.g. vulnerability curves) may lead to better results than the use of vulnerability indicators. However, vulnerability indicators are very often used for the assessment of

social vulnerability as for example in the Social Vulnerability Index of Cutter *et al.(2003)*. In the case of physical vulnerability, although vulnerability curves are more popular than vulnerability indicators, studies using indicators for assessing physical vulnerability are on the rise. Often, some of these studies focus on the development of inventories of elements at risk and their characteristics (Papathoma-Köhle et al., 2007) or  they make an additional step towards exposure assessment (Fuchs et al., 2015). Barroca *et al.* (2006) developed a vulnerability analysis tool for floods based on a system of indicators describing the

elements at risks, their spatial relation, the prevention, emergency and reconstruction systems. Barroca *et al.* (2006) suggest that the vulnerability assessment tool is simple and flexible and can be used by different users without requiring expert knowledge for its use. On the other hand, Müller et al. (2011) assessed urban vulnerability towards flood using indicators for physical and social vulnerability. Concerning physical vulnerability, four indicators were selected (construction material, building position, proportion of green spaces and local structural protection) and ranked based on expert judgement. Most of

the indicator-based methodologies found in the literature are applied at local scale. The study of (Balica et al., 2009) is noteworthy because the selected indicators used to develop indices for flood vulnerability are available for three scales (river-basin, sub-catchment and urban area). Balica *et al.* (2009) suggest that the indicator-based methodology is powerful because it supports the decision makers in prioritization and it supports transparency. Apart from floods, physical vulnerability has been investigated with means of indicators also for other hazard types such as landslides. Silva and Pereira (2014) assess

physical vulnerability of buildings to landslides based on the building resistance and the landslide magnitude. The building's resistance is determined by a number of indicators including construction technique and material, number of floors, floor and roof structure and conservation status. Last but not least Kappes *et al* (2012) developed a new indicator-based methodology for multi-hazard in mountain areas which is going to be presented in detail and applied in the present study. In general, in most of the studies found in the literature using indicators, the indicators are weighted empirically and no validation of the indicators

selection and weighting has been implemented using the damage pattern and loss from a real event.

        2.2 The PTVA method and its evolution

One of the first attempts to use indicators for the assessment of physical vulnerability was made by Papathoma and Dominey-

Howes (2003)  and Papathoma-Köhle *et al* (2003). Specifically, they developed a methodology for assessing the physical vulnerability of buildings to tsunami at coastal areas in Greece using indicators. The main concept of the Papathoma Tsunami Vulnerability Assessment Model (PTVA) was the combination of an inundation scenario with "attributes relating to the design, condition and surroundings of the building" (Tarbotton et al., 2012). The methodology was based on the fact that two buildings located exactly at the same place despite experiencing the same process intensity do not always suffer the same loss. The



reason for this is the variety of building characteristics concerning the building itself and its surroundings. A number of buildings characteristics related to the damage pattern following a tsunami event were selected (indicators) and a GIS database with the buildings and their attributes was created. The indicators were weighted using expert judgement and a vulnerability index was given to each building. The result was a series of maps for a number of coastal segments in Greece showing the spatial pattern of the relative physical vulnerability of the buildings. In this way, authorities and emergency services could focus their limited resources in specific buildings rather than to the whole potentially inundated area. The method was later validated (Dominey-Howes and Papathoma 2007) using data from Maldives following the 2004 Indian Ocean tsunami. The validation showed that the selected vulnerability indicators correlate well with the severity of the damage, however, recommendations for improvement led to an improved version of the method: PTVA-2. PTVA-2 was used in USA (Dominey-Howes et al., 2010) for the estimation of the probable maximum loss from a Cascadia tsunami in Oregon (USA). The main difference to PTVA-1 was the inclusion of the water depth above ground as an attribute in the calculation of the overall vulnerability. In this way, the method became more intensity dependant than before. PTVA-3 was developed by Dall' Osso *et al* (2009a) and was tested in Australia (Dall'Osso et al., 2009b) and in Italy (Dall'Osso et al., 2010). PTVA-3 made a step towards and more reliable weighting of the vulnerability indicators by using Analytical Hierarchy Process (AHP) rather than expert judgement to weight the attributes.

The first attempt to use vulnerability indicators for mountain hazards was made by (Papathoma-Köhle et al., 2007) with the development of an element at risk database containing indicators regarding the physical vulnerability of buildings to landslides. However, due to the lack of data regarding the hazardous process itself and the limited availability of data regarding the building characteristics the study constitutes only a good basis for further research. The next attempt was made by Kappes *et al.* (2012) introducing a methodology for vulnerability assessment using vulnerability indicators, e.g. characteristics of the building that are responsible for its susceptibility to damage and loss due to mountain hazards, based on the PTVA model. The methodology is presented in the following chapter. It is clear that as far as mountain hazards are concerned, there is definitely a need to improve and modify indicator-based approaches in order to be used for vulnerability assessment and as basis for risk reduction strategies.

The two methodological concepts have been always used in conflict rather than in combination from scientists and practitioners. A first effort to combine them has been made by (Papathoma-Köhle et al., 2015). Papathoma-Köhle *et al.* (2015) developed a tool which uses the vulnerability curve presented in the following chapters as a core to implement three functions: updating and improvement of the curve with data form new events, damage and loss assessment for future events, damage documentation of new events. However, the tool has the possibility to include information regarding building characteristics. This provides the opportunity to investigate the correlation of damage patterns and the building characteristics in the future.





## 3. The indicator based methodology: a PTVA for debris flow

The concept of the indicator-based methodology for mountain hazards (debris flow, landslides and floods) is based on the assignment of weights to a number of building characteristics resulting to a Relative Vulnerability Index (RVI) per building (Figure 1). The RVI is calculated per building by using the following equation (1):

$$RVI = \sum_1^m w_m \cdot I_m s_n \ , \qquad (1)$$

Where $w$ represents the different weights, $I$ the indicators and $s$ the sores of the indicators as shown in Figure 1. A selection of vulnerability indicators of buildings for debris flow events and the weighting according to different users are shown in Figure 2. In more detail, the vulnerability indicators are related to building characteristics (material, condition, number of floors, building surroundings) and location (row towards torrent and row towards the slope). The indicators are weighted and

10 aggregated according to equation (1) to a Relative Vulnerability Index (RVI) which is attributed to each building.

However, the weighting is not static since the needs of the end-users may vary. Therefore, each user of the method should be able to set their own priorities and change the weighting according to them. For example, emergency services need to know the physical vulnerability of the buildings in order to locate concentrated number of casualties and potential victims following

15 an event. Therefore, buildings not offering vertical evacuation opportunities (one floor buildings) are of higher importance. On the other hand, if the aim of the study is the development of vulnerability reduction efforts during the preparedness phase, building material and condition or the existence of building surroundings are of the highest importance. In this way, several RVIs may be calculated for each building: for example, an $RVI_{RP,}$ which can be used for emergency planning, or an $RVI_{VR}$ which can be used as a base for vulnerability reduction strategies (e.g. reinforcement of buildings).



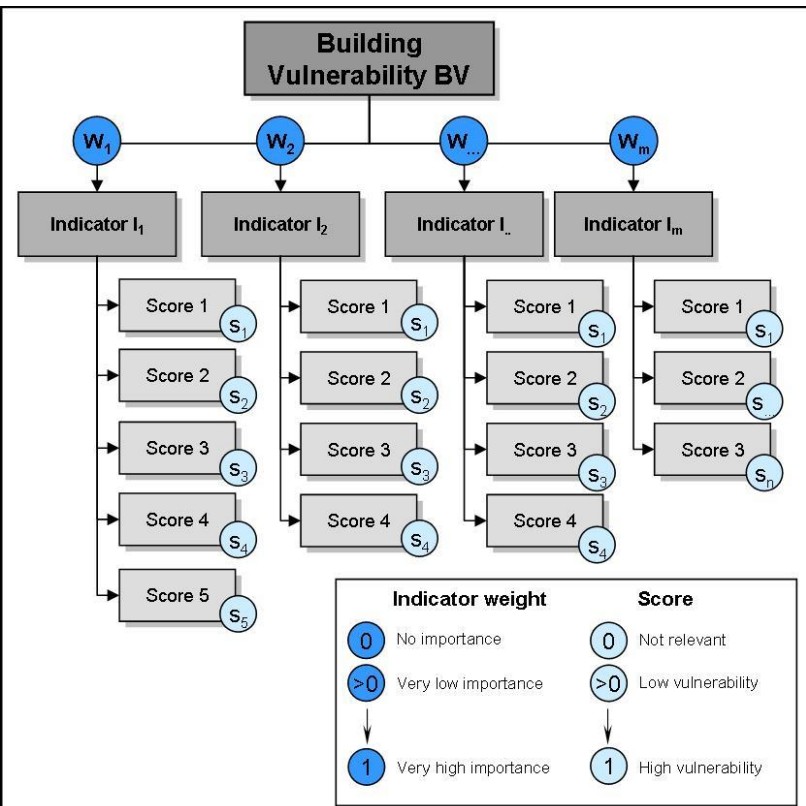

**Figure 1** The concept of the indicator-based methodology (Kappes et al., 2012)

Kappes *et al.* (2012) suggest that the main advantage of the method is the flexibility in weighting. Moreover, they consider the

5 fact that the method is not hazard intensity specific to be an advantage, because the assessment may be carried out in absence of information regarding the process characteristics. The approach is based on a "relative" vulnerability index which more or less highlights the buildings that are more vulnerable than the others. Kappes *et al* (2012) based on the limitations of the method outline the necessary actions that have to be taken in the future in order to improve the method. They recommend that damages of events have to be recorded in a more detailed way in order to comprehend the role of each indicator and their

importance in determining the vulnerability of the building. Since the method requires a large amount of detailed data alternative data collection methods may be introduced, e.g. questionnaires, google street view etc. Additional data, such data regarding the open spaces, the accumulation of movable objects or even additional elements at risk such as agricultural spaces and industry, will make the method more integrated. Furthermore, the database could be enriched with socio-economic data at building level. Last but not least, Kappes *et al.* (2012) suggest that an interesting development of the method would be the

validation of the indicators weighting based on damage records of past events.





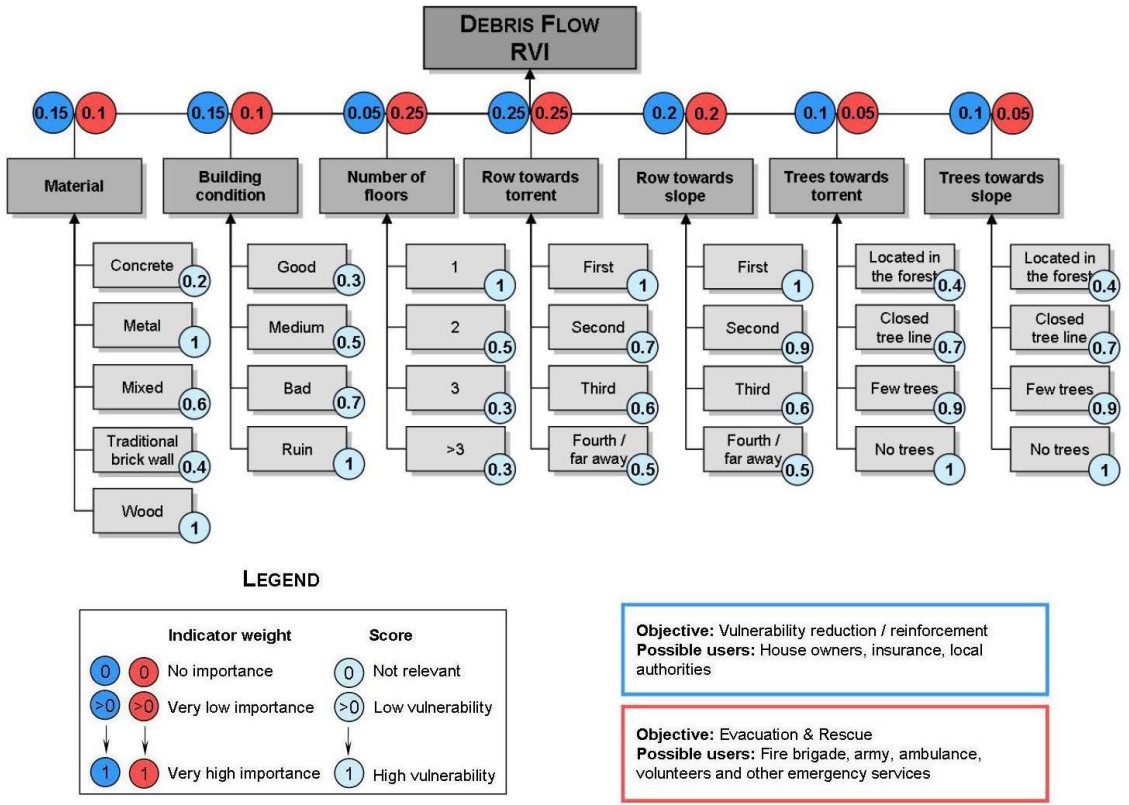

**Figure 2** The indicator-based methodology adapted for debris flow. The vulnerability indicators are demonstrated together with the weight index which varies according to the objective of the vulnerability assessment and the end-users (Kappes et al., 2012).

## 4. The case study area

The indicator based methodology has to be tested in an area that not only has a long record of debris flow events but also detailed documented damages on buildings that may reveal information regarding the intensity of the event itself but also the damage pattern on the built environment. For this reason, the methodology will be validated in Martell (South Tyrol, Italy). Communities in South Tyrol (Italy) often suffered material damages due to geomorphological hazards such as landslides, flash floods and debris flows.

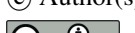


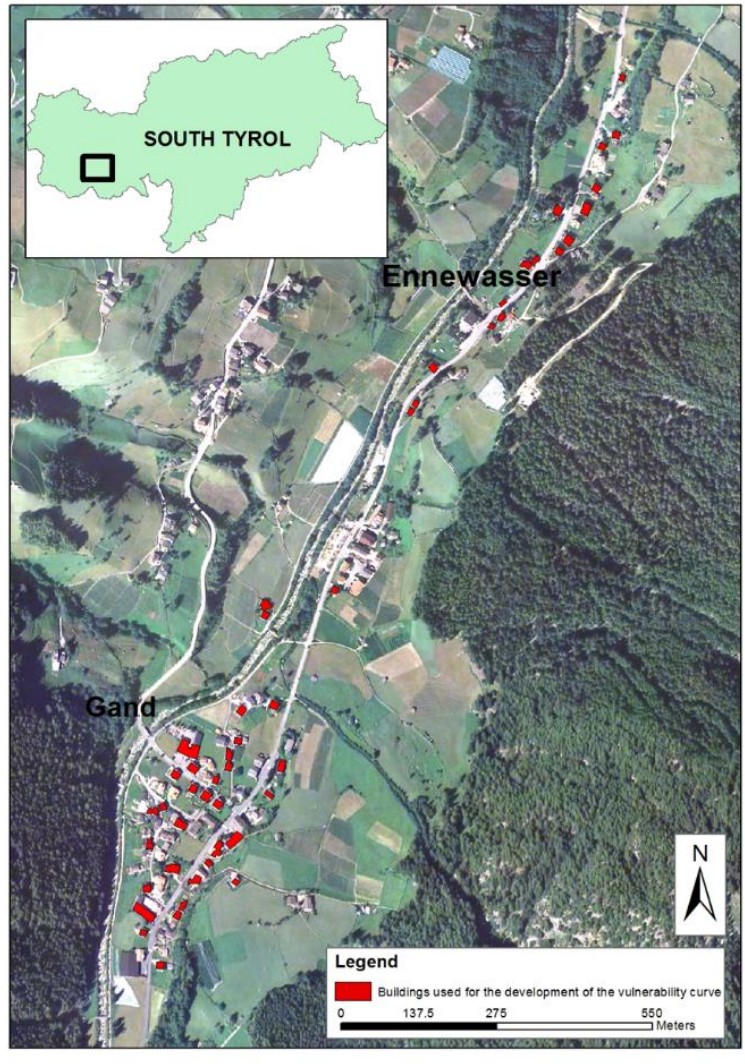

**Figure 3** The case study area: Municipality of Martell (villages of Gand and Ennewasser). The buildings used in the case study are highlighted in red colour

5   The municipality of Martell is located in the tributary valley of Vinschgau in South Tyrol, Italy. The valley of Martell is 27km long with a ranging altitude from 950 to 3700m (Martell, 2013). The settlements are located at the bottom of the valley which is mainly used for agriculture. Most of the built up areas, such as Meiern, Gand, Ennewasser and Burgaun are situated in the north part of the valley. Martell has a long record of water related natural hazard events such as glacial lake outburst, floods, debris flows and avalanches. The loose material (debris) that was left behind by glaciers during the Holocene retreat has been

10   often transported downstream by debris flows in the past causing considerable material damage to the settlements of the valley.



Additionally, a reservoir dam was constructed in 1956, which served mainly as an electrical power source and protected the village from unexpected excessive flooding.

On 24 August 1987, following some days of continuous and heavy precipitation, the river Plima transported a significantly
higher amount of water than usual. Debris flows were initiated in tributary streams along the valley. In the evening, the inhabitants were successfully evacuated, as the water level continued to rise. A couple of hours after the evacuation, a large debris flow went through the valley, causing devastation. The specific debris flow event cannot be considered as entirely natural, as it was directly connected to the mismanagement of the reservoir dam which failed to regulate the water flow into the valley (Pfitscher, 1996). The actual outflow has been estimated to be three times the usual discharge (300 to 350m3/s). The
debris flow reached the village of Gand, overflowed the river bed and found its way through the settlements. Not only buildings were destroyed, but significant damages were recorded in agricultural areas were flooded, infrastructure and the industrial zone in Vinschgau (Pfitscher, 1996). Fortunately, due to the early warning and evacuation no casualties were recorded. Although the total damage summed up to 45 to 50 billion lire (approximately 23.2 to 25.8 million EUR), private households suffered damages of slightly less than 8 million EUR.

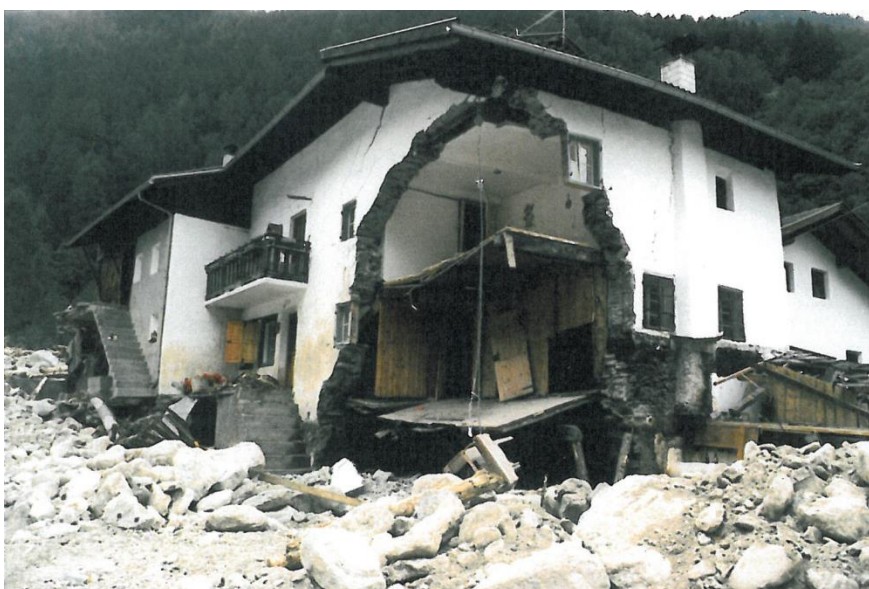

**Figure 4** Example of photographic documentation following the August 1987 event (Source: Municipality of Martell)

As far as the infrastructure, local light industry, forestry, agriculture, tourism and emergency response are concerned, damages up to 4 million EUR were recorded. Furthermore, 5.6 million EUR had to be spent for the recovery of the regional road and
the telecommunications network as well as for torrent control measures. At the time only the direct costs of the event were assessed (Pfitscher, 1996). The event was documented mainly through photographic material (Figure 4) that enabled later on the assessment of the intensity of the process as well as the damage pattern on individual buildings.

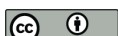



## 5. The development of a vulnerability curve for the study area

A vulnerability curve was developed based on empirical damage data of buildings in Martel, South Tyrol, Italy, that were damaged during the 1987 event (Figure 5). The event was not thoroughly documented, however, following the event the buildings, as well as parts of the affected villages, were photographed. The photographs give a very good overview of the

damage pattern, however, information regarding the costs of the damages per building, or complete description of the damage were not available. In more detail, photographic documentation of 51 buildings out of the 69 buildings that were damaged or completely destroyed during the event was used (Pfitscher, 1996), because only for this amount of buildings adequate photographic documentation was available. Based on these photos, the height of the debris deposits per building could be estimated and the monetary damage was calculated. For example, in case the damage is limited to the exterior of the building,

the works will include cleaning, plastering and painting of the external walls. In case the debris has entered the building through openings or wall breaks, there will be additional renovation works such as removal of debris and cleaning of rooms, restoration of interior walls and floor, testing of the electricity and heating network etc. (Papathoma-Köhle et al., 2012). The extent of the damage per building was translated in monetary loss based on standard prices for renovation works (Kaswalder, 2009). By comparing the value of a building in terms of reconstruction costs to the monetary damage caused by the event the degree of

loss per building could be also assessed.

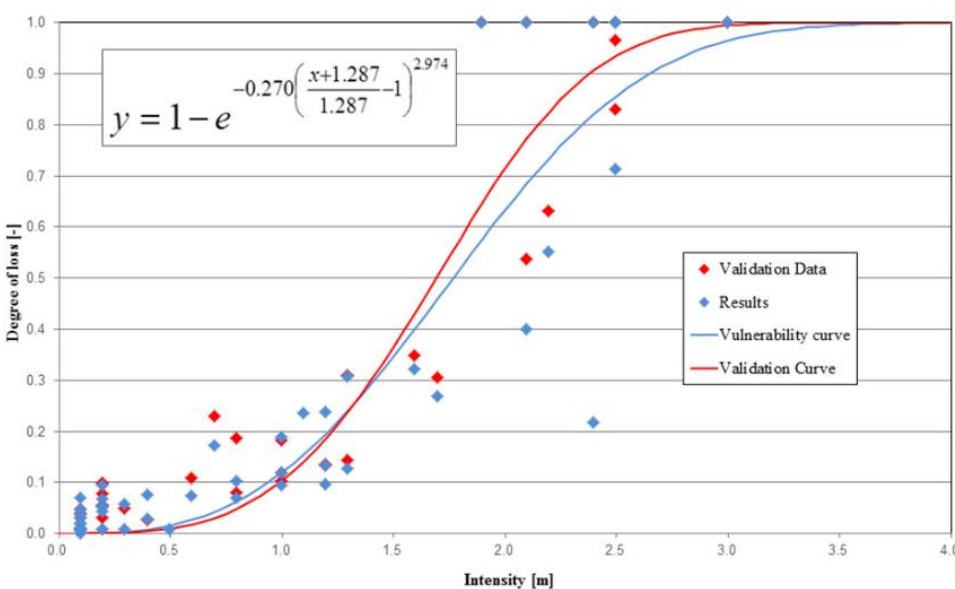

**Figure 5** The vulnerability curve and the validation curve based on damage data from the 1987 debris flow event in Martell (South Tyrol) (Papathoma-Köhle *et al.*, 2012).

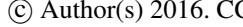



The vulnerability curve clearly shows that the higher the intensity of the process the greater the damage that an element at risk suffers. The curve indicates that if the intensity exceeds 1.5m, the degree of loss increases considerably. This may be explained by the fact that debris flow of this height may easily enter the building from windows and doors and cause additional damage in the interior. However, at this point, it is important to emphasise that only the structural damage was considered in the calculation of the monetary loss and not the content of the buildings.

Papathoma-Köhle *et al.* (2012) computed additionally a validation curve (blue curve in Figure 3) using real compensation data for only part of the buildings provided by the Department of Domestic Construction of the Autonomous Province of Bozen/Bolzano for the calculation of the degree of loss. The visual comparison of the two curves demonstrated the validity of the developed vulnerability curve.

The results (intensity and degree of loss) are displayed herein for the first time in two separate maps (Figure 6-intenisty and Figure 7-degree of loss) in order to demonstrate the spatial pattern of the two factors. By the maps it is clear that buildings located very close to the steep slope experienced higher intensity. For buildings situated next to the road also high intensity values were recorded, probably because roads acted as corridors that enabled the debris flow to enter the settlement area. Moreover, in Figure 6 is also obvious that in Ennewasser (north of the map) the intensities were significantly lower due to the protection offered by the forest on the east side of the settlement. In Figure 7 the spatial distribution of the degree of loss follows in most cases the pattern of the intensity distribution. As Table 1 clearly shows the majority of the buildings experienced rather low intensity of debris flow (less than 1m debris height). Only 9 buildings (17,5%) experienced high intensity of debris flow (more than 2m debris height).

**Table 1** Intensity categories and the corresponding number of affected buildings

| Intensity | Number of buildings | % |
|---|---|---|
| <1 | 30 | 59 |
| 1-2 | 12 | 23,5 |
| >2 | 9 | 17,5 |

Observations based on the vulnerability curve may also lead to the selection of indicators e.g. height of windows, proximity to the road, importance of surrounding vegetation.





**Figure 6** The spatial pattern of the assessed intensity of the 1987 event per building in the villages Gand and Ennewasser.



**Figure 7** The spatial distribution of the assessed degree of loss per building following the 1987 debris flow event in Gand and Ennewasser.




## 6. The application of the indicator-based methodology in the study area

The indicator-based methodology was applied at each building that experienced damage during the event of 1987 in Gand and Ennewasser (Municipality of Martell). Based on photographic documentation provided form the municipality of Martell a GIS a database was developed containing the vulnerability indicators of each building. Information regarding the material, the building condition and the intensity could be collected from photos, whereas, information regarding the building surroundings and the building's location in relation to the neighbouring buildings could be acquired from an ortho-photo of the area. An example of the calculation of physical vulnerability using indicators is shown in Figure 8. The photo shows a building which was damaged from the debris flow in 1987. The intensity of the process and the degree of loss have been assessed and calculated respectively during the development of the vulnerability curve (Papathoma-Köhle et al., 2012).

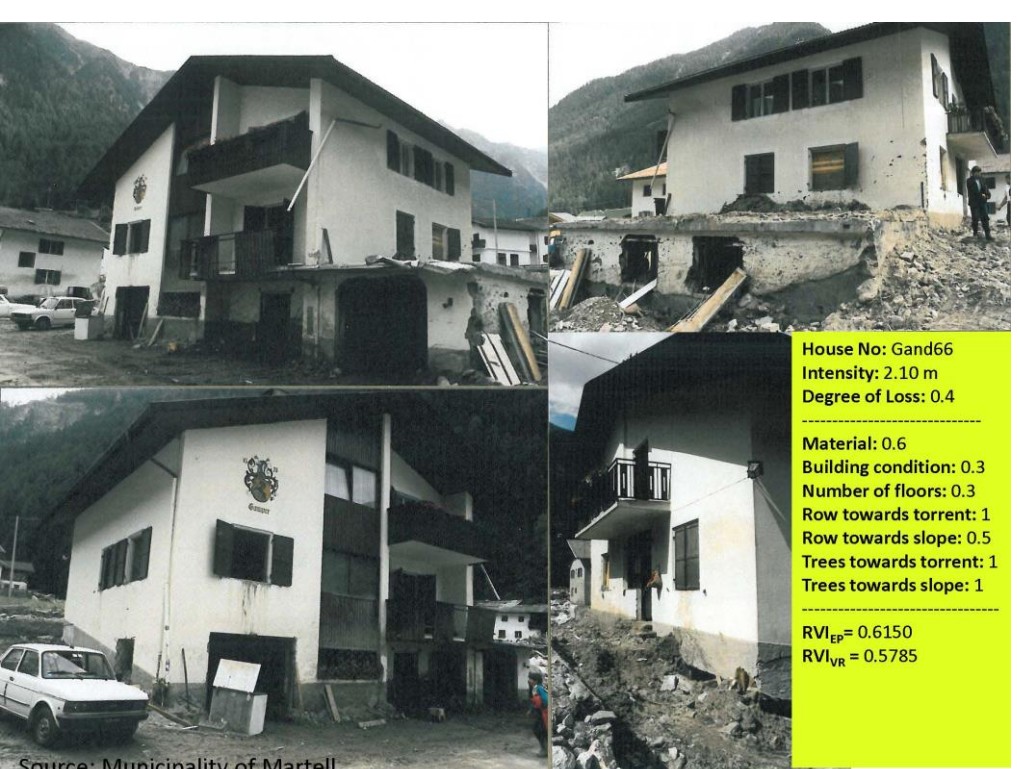

**Figure 8** An example of the application of the indicator based methodology for two different objectives: for evacuation planning (RVI$_{EP}$) and for vulnerability reduction strategies and reinforcement (RVI$_{VR}$).

The vulnerability indicators were collected for each building as it is shown in the example of Figure 8 an RVI$_{EP}$ and RVI$_{VR}$ was calculated. The application of the RVI at Martell shows that the weighting of the vulnerability indicators by different end-users leads (at least in this case) to rather similar results (Fig. 9 and 10 and Table 2) and that differences are focused on individual buildings.



**Figure 9** The spatial distribution of physical vulnerability using indicators for emergency planners





**Figure 10** The spatial distribution of vulnerability based on indicators weighted for vulnerability reduction strategies



In the following paragraphs the results of the two methods are compared, the benefits and drawbacks of the indicators based method are listed and improvements not only for the further development of indicator-based methods but for the assessment of the physical vulnerability in general are outlined.

## 7. Discussion

7.1 Comparison of the results of both methods

Damage and intensity data that are used for the development of vulnerability curves are rarely displayed on a map. The curves are only a tool that may be used to predict the degree of loss of a building should it experience a specific intensity. However, in the present article the spatial pattern of the intensity and the degree of loss are displayed in two maps in Figure 6 and 7
respectively. The comparison of the results of the indicator-based methodology (Figure 9 and 10) with the intensity of the process per building (Figure 6) of the 1987 event and the spatial distribution of the degree of loss (Figure 7) (Papathoma-Köhle et al., 2012) shows that the results of both methods are compatible. The buildings that experienced a high degree of loss (Figure 7) are often the ones with a high RVI especially in the case of $RVI_{VR}$. However, the visual comparison of the maps shows that some buildings that experienced a very high degree of loss have been assigned with a low vulnerability index and vice versa.
For example, the buildings in Ennewasser, that have experienced low intensities and for this reason they also showed a relative lower degree of loss, were classified as highly vulnerable by the indicator based method. This observation highlights an important aspect of the indicator-based method which is the fact that the indicator based methodology is not hazard intensity specific in contrast to the vulnerability curve method that includes information regarding the intensity. On the other hand, there are buildings in Gand that although they have experienced a high degree of loss they are assigned with a low RVI. The high
degree of loss it is expected since according to Figure 6 the same building also experienced high intensity. It is, therefore, clear the results show inconsistencies because the intensity is not considered by the indicator-based methodology. However, there are building characteristics that are directly connected to the intensity that a building experiences e.g. surrounding vegetation or protection from other buildings, proximity to the road network etc.

By visualising the spatial pattern of the degree of loss and the observed intensity, additional valuable information regarding the importance of vulnerability indicators may become available and, in this way, the indicator-based methodology may be modified, extended and substantially improved. For example, the low intensities of recorded on buildings in Ennewasser may be related to the presence of surrounding vegetation which highlights the importance of the relevant vulnerability indicator. This leads to the conclusion that both methods, although based on a different concept they shed light in two different aspects
of the physical vulnerability, namely the characteristics of the buildings and the expected degree of loss under a given intensity respectively and, therefore, they should be used in combination rather than in conflict.
However, additionally to the visual comparison of the results, a closer look to the results for each building in Table 2 reveals even more about the advantages and disadvantage of both methods.



**Table 2** Comparison of the results per building. The colours indicate the degree of intensity as described in Table 1. (yellow for Intensity <1m, orange for intensities 1-2m and red for intensities>2m)

| Building ID | Intensity | Degree of loss | RVI$_{VR}$ | RVI$_{EP}$ | Building ID | Intensity | Degree of loss | RVI$_{VR}$ | RVI$_{EP}$ |
|---|---|---|---|---|---|---|---|---|---|
| G 26 | 2,10 | 1,00 | 0,6565 | 0,88 | G 226 | 2,40 | 0,22 | 0,5055 | 0,565 |
| G 28 | 1,20 | 0,10 | 0,6285 | 0,735 | G 41 | 0,10 | 0,04 | 0,4335 | 0,53 |
| G 29 | 2,40 | 1,00 | 0,6615 | 0,76 | G 219 | 0,10 | 0,02 | 0,5305 | 0,55 |
| G 30 | 0,80 | 0,10 | 0,5885 | 0,665 | G 42 | 0,10 | 0,02 | 0,4305 | 0,51 |
| G 32 | 2,50 | 0,71 | 0,6135 | 0,69 | G 45 | 0,20 | 0,05 | 0,4665 | 0,56 |
| G 35 | 1,90 | 1,00 | 0,6285 | 0,735 | G 46 | 0,30 | 0,06 | 0,5605 | 0,62 |
| G 36 | 0,20 | 0,07 | 0,5855 | 0,645 | G 60 | 0,20 | 0,01 | 0,5335 | 0,61 |
| G 37 | 0,60 | 0,07 | 0,5855 | 0,645 | G 62 | 0,10 | 0,03 | 0,6035 | 0,64 |
| G 38 | 0,10 | 0,00 | 0,6815 | 0,735 | M 68 | 0,10 | 0,05 | 0,6755 | 0,695 |
| G 224 | 1,60 | 0,32 | 0,5985 | 0,71 | M 69 | 0,10 | 0,01 | 0,6945 | 0,805 |
| G 47 | 0,10 | 0,01 | 0,6785 | 0,715 | E 148 | 0,50 | 0,01 | 0,6755 | 0,74 |
| G 48 | 0,10 | 0,01 | 0,6315 | 0,755 | E 153 | 0,10 | 0,01 | 0,6485 | 0,7 |
| G 49 | 0,10 | 0,01 | 0,6215 | 0,705 | E 154 | 1,30 | 0,31 | 0,6755 | 0,74 |
| G 55 | 1,10 | 0,24 | 0,6945 | 0,805 | E 157 | 0,10 | 0,01 | 0,5845 | 0,655 |
| G 54 | 0,20 | 0,04 | 0,5905 | 0,65 | E 159 | 0,20 | 0,01 | 0,6355 | 0,765 |
| G 225 | 0,10 | 0,01 | 0,6855 | 0,745 | E 160 | 0,30 | 0,01 | 0,6855 | 0,745 |
| G 52 | 1,00 | 0,19 | 0,5565 | 0,65 | E 161 | 1,00 | 0,12 | 0,6755 | 0,695 |
| G 53 | 1,70 | 0,27 | 0,5145 | 0,625 | E 162 | 0,40 | 0,08 | 0,6455 | 0,68 |
| G 56 | 3,00 | 1,00 | 0,5785 | 0,615 | E 221/A | 0,20 | 0,09 | 0,6885 | 0,765 |
| G 57 | 1,20 | 0,24 | 0,4885 | 0,565 | E 221/B | 0,20 | 0,05 | 0,6755 | 0,695 |
| G 59 | 2,50 | 1,00 | 0,6685 | 0,745 | E 170 | 0,80 | 0,07 | 0,5385 | 0,635 |
| G 58 | 2,50 | 1,00 | 0,5785 | 0,615 | E 171 | 0,70 | 0,17 | 0,6885 | 0,765 |
| G 66 | 2,10 | 0,40 | 0,5785 | 0,615 | E 172 | 1,00 | 0,10 | 0,6755 | 0,695 |
| G 65 | 1,30 | 0,13 | 0,6085 | 0,685 | E 173 | 0,40 | 0,03 | 0,6655 | 0,69 |
| G 64 | 2,20 | 0,55 | 0,5035 | 0,54 | E 176 | 0,10 | 0,07 | 0,6585 | 0,75 |
| G 63 | 1,20 | 0,13 | 0,4885 | 0,585 | | | | | |

In Table 2 the results of both methods are displayed for each of the 51 buildings of the case study. The comparison of the results lead to a number of interesting observations:



1.  The value of the RVI vary from 0,43 to 0,88. No extreme values are observed. There are no buildings with RVI=1. This is to be expected since coincidently there are no buildings in the area with extreme scores (e.g. material=wood or condition=ruin).

2.  There are no buildings with RVI=0. That would mean that a building is not vulnerable. However, the assignment of zero scores that could lead in such a result is not possible. This is understood since no building can be characterised by zero vulnerability since it is located within the area affected by the debris flow. This is also supported by the vulnerability curve which is based on a real event and shows that no building had zero degree of loss.

3.  As far as the comparison between the two methodologies is concerned, some buildings although they have experienced low degree of loss and they are assigned with a high RVI. This can be explained by the fact that the indicator based methodology is not intensity specific. The high RVI means that the specific building could experience high degree of loss due to its characteristics. On the other hand, the vulnerability curve shows that the specific building did not experience high degree of loss during the specific event for reasons that maybe connected to characteristics of the process itself (e.g. G48). This is even more obvious in the case of buildings G53 and G56. The two buildings have been assigned with similar RVIs. They have, however, experienced degree of loss 1 and 0,27 respectively. This can be explained by the significant difference in the process intensity (1,7m and 3m respectively).

4.  However, the opposite observation is also clear: some buildings have experienced high intensity which resulted in a low degree of loss (e.g. G53, G64, G66, G224, G226). Obviously the fact that buildings although they experience the same intensity of the process they experience different degree of loss (some are completely destroyed and some others have less than 50% of loss may be explained by differences in their characteristics. However, another explanation may be characteristics of the process that are not considered by the vulnerability curve such as velocity of the flow, direction of impact, viscosity, duration, size of debris etc.

The results of the indicator methodology depend on the set of indicators used for the calculation of the RVI. The indicator-based methodology is based on a set of indicators that were selected based on expert judgement, damage reports and photographic documentation of past events. However, Birkmann (2006) based on existing set of indicators (EEA, NZ Statistics etc.) provides a list with quality criteria that vulnerability indicators have to fulfil. In Table 3 the set of indicators of the present study is tested towards some of these criteria.

**Table 3** Quality criteria for vulnerability indicators (adapted from Birkmann (2006))

| Criteria | Vulnerability indicators |
|---|---|
| Measurable | The indicators used are not always easily measurable. The difference between a building of "good" or "medium" condition is not clear and not measurable in quantitative terms. The scores for each building may be dependent on the judgement of the data collector and may not always be objective. However, |




| | improved data collection techniques (e.g. detailed standardised questionnaires etc.) may improve the measurability of the indicators. |
|---|---|
| Relevant | The indicators have been based on reports and documentation of past events and for this reason are relevant to the assessment. They have also been chosen according to the needs of the end-users although the latter could be more involved in the selection process in the future. The weighting, however, is done directly by the end-users. |
| Policy-relevant | Although not demonstrated in the specific article, the indicators may be policy relevant. The vulnerability indicators may give decisions makers an overview of damage potential for future events. Moreover, they may be used for emergency planning and they may guide local structural protection measures. |
| Measure important | The indicators are connected to key elements (e.g. reaction of the structure to the impact of debris flow) and are not attempting to indicate all aspects (e.g. vulnerability to other hazard types). |
| Analytically and statistically sound | Although the indicators are analytically sound in the sense that they may depict the actual situation with accuracy, the links between natural process and degree of loss as well as the reaction of a structure to the natural process according to its characteristics are not fully understood and therefore, further research is required. |
| Understandable / Easy to interpret | The indicators used in this study are easy to interpret. No expert stuff is required for their collection. Although this is an advantage of the method, the judgement of the collector may influence the result significantly. |
| Sensitivity | Although the indicators are specific to the phenomenon of debris flow, they are not sensitive in changes of this phenomenon such as its intensity. However, they are sensitive to changes in the structure of the building which means that they are able to express changes in the physical vulnerability should a building is reinforced. |
| Validity/accuracy | The indicators have the capacity to express the physical vulnerability of buildings in most of the cases and this may be also confirmed by the results of the vulnerability curve. However, this is not always the case. The cases where high vulnerability has been assigned for buildings that have experienced low degree of loss have to be investigated and based on the conclusions of this investigation the methodology has to be improved. |
| Reproducible | Theoretically, the indicators set could be reproducible for another area facing a threat of debris flows. However, significant differences in the architecture and building standards of buildings should be considered. |



| Based on available data | The indicators are not always based on available data. Some information could be available from the municipality. However, the majority of the required information may be collected through field work or interpretation of ortho-photos. |
|---|---|
| Data comparability | The indicators may be compared to similar ones in other areas but also to past conditions of the same area of future hypothetical ones. |
| Cost effective | The indicators are cost effective. The assessment of the vulnerability for debris flow usually involves a limited amount of buildings. Although fieldwork is necessary there are ways to avoid it by sending questionnaires to the building owners or interpreting ortho-photos for rapid data collection. |

Table 2 shows that although the set of indicators fulfils many criteria, there is still room for improvement of the indicator set itself, the description of the attributes for each indicator and the collection of the required data.

5        6.2 Benefits, limitations and future development of the indicator-based method

In contrast to the vulnerability curves, indicator-based methodologies are not established and applied by practitioners in the same dimension. For this reason, the comparison between the two concepts is essential in order to highlight the benefits of the indicator-based methodologies, point out its weaknesses and make a list of recommendations not only for the improvement of
the indicator-based concept but for the improvement of the assessment of physical vulnerability in the future.
The comparison of the two methods highlighted the following benefits of using indicators for assessing physical vulnerability of buildings to torrential hazards:

1. In contrast to the development of vulnerability curves, the indicator based methodology prompt the user to develop an **inventory of the elements at risk** and a database of building characteristics. This may enable the development
of strategies for the reduction of vulnerability at a very local level, as well as the development of local structural protection measures. In this way the focus and the resources will be concentrated on limited amount of buildings.

2. The indicator-based methodology does not use empirical loss data but it rather indicates the **relative vulnerability** of individual buildings. This means that this type of methodologies can be applied to areas with no recorded history of events making their use easier in absence of damage data.

3. The **weighting is flexible** and may be adjusted to the needs of individual users. This ensures the use of the methodology by a range of end -users.

4. **No experts** are required for the data collection. The assignment of the scores of the individual indicators do not require any expert knowledge. This means that even the owners of the buildings may provide the required information themselves saving in this way money and time for data collection.

5. The use of GIS makes the **updating of information** easier but also by changing the scores of indicators we can answer "what if" questions related to local structural protection and reinforcement of buildings. Moreover, by using



easy to update databases future changes in the spatial pattern of the built environment, socio economic and land use changes may be considered for future scenarios.

6. The use of GIS enables the **visualisation** of the spatial pattern of the physical vulnerability and also individual characteristics. In this way vulnerability maps may be used as a basis for e.g. emergency planning. On the contrary, vulnerability curves do not have a spatial component.

7. The **transferability** of methods in the field of risk research is often due to differences in the nature of elements at risk, environmental conditions and processes impossible. However, since the specific method is not intensity and process dependant, it is easier to transfer to other areas with similar problem provided that necessary modifications will be made (e.g. additional building characteristics due to local architecture).

8. Indicator-based methods encourage the **involvement of local communities** and individual building owners in data collection and vulnerability reduction.

However, indicator-based methodologies for physical vulnerability assessment are in their infancy allowing a large space for improvement. This improvement requires the identification of the main drawbacks of the methodology:

1. **Intensity relevance:** the indicator methodology assigns a relevant vulnerability index to each building which more or less shows which building is more vulnerable than the other in a worse-case scenario without indicating a specific intensity of the event. This is a major drawback which is obvious from Table 2: buildings with high RVI experienced a very low degree of loss. A possible development of the method could be similar to the one of PTVA-2 which included the water height (in this case the deposit height) as an indicator in the vulnerability assessment, based on a specific scenario.

2. **Completeness of the set of indicators:** From Table 2 is obvious that there some inconsistencies between the two methods for specific buildings. The reason may be the lack of completeness of the set of indicators. For example, characteristics that may affect the vulnerability of the building significantly, such as the existence of openings on the slope side, their size and quality, as well as the existence of a basement are not considered.

3. **Completeness of data sets and costs of data collection:** The data set required for the implementation of the methodology is detailed and has to be collected at local level.

4. **Description of scores:** The scores for the various indicators are often described in a trivial way. E.g. "good" or "medium" building condition leading to a large dependence on the judgement of the data collector to decide the score that will be assigned to a building.

5. **Classification of results:** The classification of the results may also change the overview of the spatial pattern of the values. The classification method used in the present study was the "Equal Interval Classification Method" whereas, Kappes *et al* (2012) used the "Quantile Classification methods" arguing that in this way the users may set priorities. In any case the classification method as well as the weighting should be also decided by the user.



6.  **"Relativeness" of the vulnerability indicator:** The RVI expresses the relative vulnerability of buildings in an area. That means that the RVI points out which building is more vulnerable than the other without having the capacity to translate this vulnerability into a quantitative value. This may be considered a disadvantage for practitioners because the vulnerability map made this way may only indicate vulnerable buildings in an abstract way and vulnerability maps from different areas may not be compared.

7.  **Uncertainties:** Both methods bear a number of uncertainties that should be analysed and quantified. As far as the end-users are concern, decision makers are in need of vulnerability assessment methods that they can use for risk analysis but they also need to know the uncertainties that are associated with the vulnerability values. This is important because low estimated risk with high uncertainties may cause higher losses than medium estimated risks with minor uncertainties. (Papathoma-Köhle et al., 2012) lists the sources of uncertainties that are related to the development of the vulnerability curve:

    a.  Intensity of the event: attributes of the intensity of the process such as duration, velocity, direction, etc. are ignored.

    b.  Damage pattern: photographic documentation has been used for the identification of the damage pattern. Information regarding the damage of the interior of the buildings is missing.

    c.  The degree of loss was based on the assessment of the cost of reconstruction and the value of the building. Both assessments bear a significant number of uncertainties (e.g. existence and size of basement changes the building value, impact on the electricity and heating network).

    d.  Credibility of existing data: Some buildings although they were not severely damaged got full compensations to be rebuilt due to relocation.

    As far as the indicator-based method is concerned, uncertainties are related to the following:

    a.  The subjectivity of the data collector (e.g. what is a "good" and what is a "medium building condition?)

    b.  The subjectivity of the end-user concerning the weighting and the classification method.

    Efforts to analyse and quantify uncertainties concerning the assessment of physical vulnerability for debris flows have been made in the past ((Totschnig and Fuchs, 2013),(Eidsvig et al., 2014)), however, most of them concern vulnerability curves.

Based on the comparison of the two methods and the outline of the advantages and disadvantages of it as far as the assessment of physical vulnerability is concerned a number of recommendations for improvement may be made:

1.  **Reduction of data collection effort and time** through the use of modern technologies (GIS and remote sensing): Information regarding the surroundings or the building row is easier to be collected by using GIS maps and/or remote sensing where needed. For other type of information such as openings, existence of basement and building material and condition an additional field survey may be necessary. Alternative data collection methods (e.g. distribution of standardised questionnaires to the building owners) may also be introduced.





2. Improvement of **post-disaster documentation** methods: Detailed post disaster documentation including photographic material may provide valuable information regarding the interaction between natural processes and elements at risk but it may also give information regarding the variation of the intensity of a process within a given area. An improved method for damage documentation is recommended by Papathoma-Köhle *et al (2015)*.

3. Reconsider the **score description**: some scores would be more reliable if they would be less dependent on expert judgment (for example building condition). A solution would be to reconsider their descriptions and come up with tangible scores (e.g. age of the building, or detailed description of building condition).

4. **Additional indicators:** The list of indicators used for the assessment of physical vulnerability of buildings to debris flow is not exhausted. Information regarding the presence of a basement, the number location, size and quality of openings is also missing. Furthermore, information regarding the local structural protection of buildings (e.g. elevation, splitting wedges, deflection walls etc.) should also be included.

5. **Physical resilience:** Some of the indicators mentioned above may also contribute not only to the assessment of the physical vulnerability of the building but also to the assessment of its physical resilience. According to Haigh (2010, p.4) resilient buildings should "enable society to continue functioning when subject to a hazard". In this respect, if the interior of the building has been invaded by debris or if the heating and electricity network are located in the basement and have been heavily damaged by intruding water and material the building will need more time to be re-inhabited by its occupants. Therefore, indicators regarding the physical resilience of buildings may include location of vital equipment, existence of openings, local protection measures, basement, primary or secondary residence of the inhabitants).

6. The interaction of structures with the natural process has not been thoroughly investigated. Gems *et al*. (2016) investigate the interaction among buildings and debris flow and they give insights of the impact of the flooding in the interior and on the exterior of the building. However, further **experiments** are needed.

7. **Improved weighting of indicators:** The weighting of indicators is based on expert judgement increasing in this way the level of uncertainty. An improved weighting could base the hierarchy of indicators on statistical analysis of their importance correlating real damage data (monetary cost of damages) to building characteristics.

8. **Application for multi-hazard:** Kappes *et al.* (2012) presented the specific methodology for multi-hazard. In their study the made three separate maps based on the indicator based methodology for flood, shallow landslide and debris-flow. However, as Kappes *et al* (2012) also pointed out, there is a need to consider how different hazard types affect the vulnerability of buildings to other hazards that may happen simultaneously or in a short time span. A database which includes indicators related to more than one hazard type would be a good start.

Improving indicator-based methodologies is an important step towards disaster risk reduction, however, the advantages of the vulnerability curves are also indisputable, a comparison of the two methods may shed light to their advantages and drawbacks and may also inform the practitioners on their available methodological choices. However, as it is suggested in the following paragraphs both methodological approaches are needed.


7.3 Vulnerability curves vs. vulnerability indicators

The above comparison of the two methodological concepts clearly shows that decision-makers and other potential end-users

are actually in need of both methods during the different phases of the disaster management cycle for different reasons. In short, decision makers, authorities and disaster managers need a quantitative representation of physical vulnerability based on empirical data (curves) as well as, relative vulnerability values for individual elements that are directly connected to their characteristics (indicators) and may support prioritisation of resources and small local interventions for vulnerability reduction. The fact that both approaches are significant stresses the need for a "holistic physical vulnerability framework for practitioners"

that will enable practitioners to choose the relevant method according to their aim and the available resources. An effort to place the various methodological concepts (including in this case also vulnerability matrices) within the phases of the disaster cycle has been made by (Papathoma-Köhle and Ciurean, 2014).

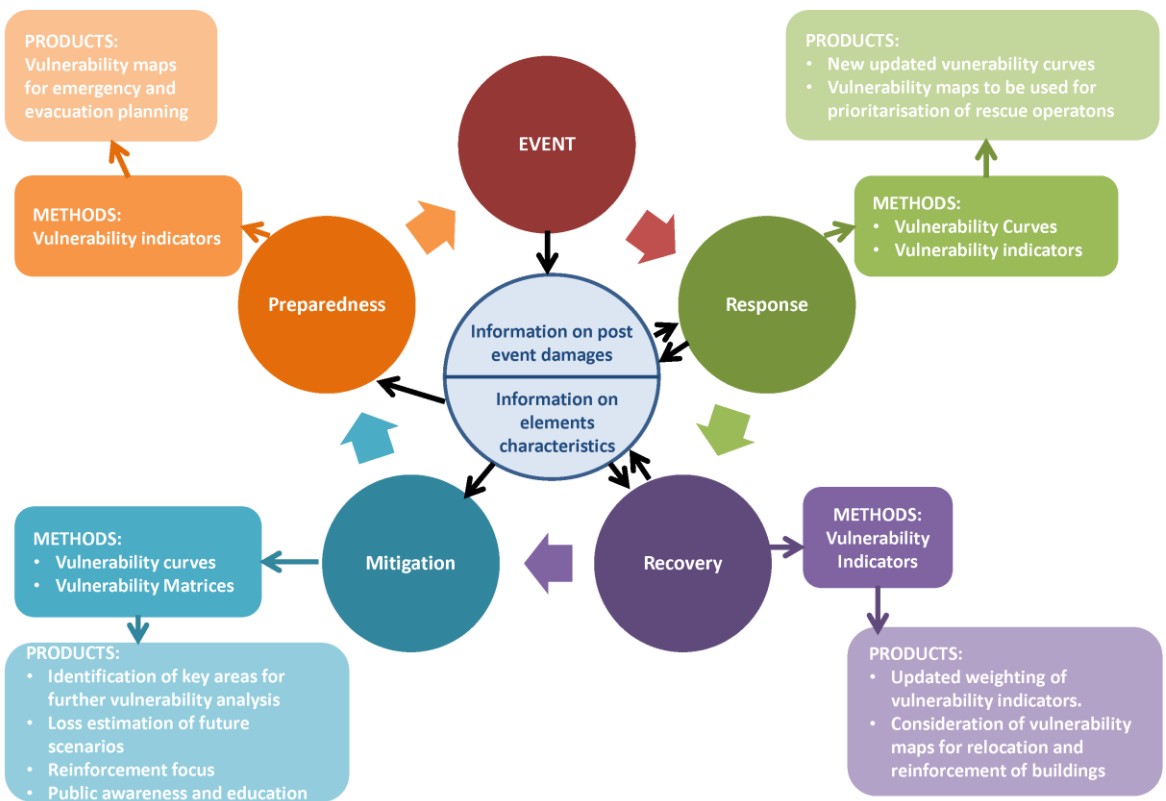

**Figure 11.** The role of vulnerability assessment and the corresponding methodological concepts in the phases of the disaster

cycle (Papathoma-Köhle and Ciurean, 2014).



Figure 11 shows the importance of vulnerability assessment in every phase of the disaster cycle. However, for each phase a different method is relevant. For example, during the mitigation phase vulnerability curves may be used for the loss estimation of future events and the design of mitigation measures. The results in this case may be used for public awareness and education

but also for recommendations for reinforcement. On the other hand, vulnerability maps based on the vulnerability indicators may guide the emergency and evacuation planning process during the preparedness phase. Additionally, during the response phase, vulnerability maps are essential to indicate the buildings that are more possible to have been damaged, whereas the recovery phase is the ideal phase for validating the weighting of indicators. Moreover, during the recovery phase the results of the vulnerability assessment may be used as guidance for relocation of buildings. Furthermore, it is essential that methods and

tools for analysing and quantifying uncertainties should be also included in a "holistic physical vulnerability framework for practitioners". The scheme in Figure 11 could form the basis of a framework for practitioners, however, further work is needed on this respect.

## 8. Conclusions

Vulnerability assessment constitutes a large part of risk analysis and its reduction has a direct effect on the consequences of natural disasters on communities, buildings and infrastructure. The variety of methods available are used until now mostly in isolation and there is often a debate about the advantages and disadvantages of different methods. In this chapter, through the comparison of two of the most common vulnerability assessment approaches (vulnerability indicators and vulnerability curves) a new methodology is applied and validated. The vulnerability curves although widely used by practitioners they reveal the

need to include the characteristics of the buildings (indicators) in the assessment of physical vulnerability. The indicator-based method presented and applied in the present article shows that although assessing physical vulnerability using indicators may give reliable results, there is significant work that has to be done in order to improve indicator-based methodologies for physical vulnerability assessment. Moreover, the article emphasizes the need for a "holistic framework for physical vulnerability assessment for practitioners" that will offer the opportunity to the end-users to use both methodologies in a complementary

way, quantify uncertainties and consider change, not only as far as the natural process is concerned but also socio-economic and land use changes that will have a direct influence on the consequences of natural disasters.

**ACKNOWLEDGMENTS**

This project received funding form the Austrian Science Fund (FWF): P 27400. Data collection was partly supported by the

EU project MOVE (Methods for the Improvement of Vulnerability Assessment in Europe) (contract number 211590). The author would like to thank the Local Authorities of Bolzano and the Municipality of Martell for the provision of data and photographic material. Furthermore, Thomas Thaler and Sven Fuchs are thanked for the discussions on earlier drafts of the present article.



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
