# Peer review of "Vulnerability curves versus vulnerability indicators: application of an indicator-based methodology for debris-flow hazards"

_Natural Hazards and Earth System Sciences, 2016_

## Referee Comment (RC1) · Anonymous Referee #1 · 15 Apr 2016

General comments:

The paper presents a useful overview and comparison of pros and cons of vulnerability assessment using vulnerability curves and vulnerability indicators. Unfortunately, the author has a tendency to cite her own work more often than is warranted. Furthermore, engineering approaches to modelling physical vulnerability are completely ignored in the paper. For example, the fact that vulnerability curves stem from the fragility curves that were developed for earthquake engineering applications when the earthquake risk assessment software package HAZUS was developed in early 1980s, is not mentioned at all.

Another example of what is not mentioned is that vulnerability curves for various types

of hazards and various building types can be developed by numerical modelling and simulations. This requires access to and experience with large non-linear finite element codes (like ABAQUS) and could be computationally intensive. However, high-performance computational capabilities are readily available to research organizations and engineers firms these days, and development of vulnerability curves by numerical modelling is becoming more and more popular for earthquakes, tsunamis, debris flows, and even ashfall from volcanic eruptions. It is quite surprising that this approach, which is relatively common in engineering applications, is not mentioned at all by the author.

Specific comments:

Page 2, lines 2 and 3: This statement is strictly not correct. Ignoring the characteristics of the building is a choice made by the person developing the vulnerability curves, mainly because of lack of data. In earthquake engineering, vulnerability curves (which are called fragility curves in that discipline) are developed for different classes and typologies of buildings.

Page2, line 13: The definition of physical vulnerability is not in "conflict" with other, more general definitions of vulnerability. It is just one possible quantitative interpretation of it.

Page 18, line 14: The author seems to forget that physical vulnerability is basically the conditional probability of loss, given that a hazard of certain occurs. A building with high vulnerability will suffer little damage if it is subjected to low intensity hazard. Likewise, a building of low vulnerability may be totally destroyed during an extreme event.

Page 18, line 21: The indicator-based relative vulnerability index could be transformed to site-specific vulnerability curves if enough data exist for doing the transformation. When there is lack of data, the transformation could be based on expert engineering judgement. This of course involves some uncertainty, but the level of uncertainty is not necessarily greater than the variability observed when data are available (e.g. Figure 5 of the paper).

---

## Author Comment (AC1) · 26 Apr 2016

Response to Anonymous Referee #1

I would like to thank the anonymous referee #1 for his remarks. Following, you can find my response to the general and the specific comments.

General Comments:

-The referee points out that the authors work has been cited too often in the article (9 out of 37 References). I considered this necessary since I have been working with both methodologies extensively in the past, however, I could certainly reduce this number in a new version of the paper. -A historical review of the vulnerability curves is

missing indeed. The author would like to confirm that she would include this review in a new version of the paper if the editor considers it also necessary. -Reference to engineering approaches such as numerical modelling and simulations is indeed limited for the following reasons: 1. The focus of the paper is not a review of all existing approaches for the vulnerability assessment but the comparison of two dominant approaches (curves and indicators). 2. The approaches that the referee refer to in the comment, in my opinion, are not vulnerability assessment approaches as such, but engineering approaches (process, impact, or structural and physical response modelling) that produce information that may be used by vulnerability assessment approaches. For example, numerical modelling of a past debris flow event may reveal information on the intensity on individual buildings that may be used for the development of a vulnerability curve (e.g. Quan-Luna et al, 2011) or simulations of debris flow events in the lab may be used to define thresholds for wall collapse under different intensities, grain size, viscosity, etc. that may be also used for the development of vulnerability curves (Gems et al., 2016, Mazzorana et al., 2014). A reference to these approaches is also possible in a reviewed version of the paper especially as they offer an alternative to empirical data for the development of vulnerability curves.

Specific Comments:

Comment: Page 2, lines 2 and 3: This statement is strictly not correct. Ignoring the characteristics of the building is a choice made by the person developing the vulnerability curves, mainly because of lack of data. In earthquake engineering, vulnerability curves (which are called fragility curves in that discipline) are developed for different classes and typologies of Buildings

Reply: I have to agree with the referee as far as earthquake hazards (and maybe also floods) are concerned. In most of the other cases (having debris flow in mind) the person developing the vulnerability curves has rarely enough buildings to make several curves for different types of buildings. Additionally, an indicator-based methodology would make use of numerous building characteristics and not only the building type

(surroundings, number of openings, presence of basement etc.). Making a set of vulnerability curves for each of these characteristics would be time consuming. However, I agree that the statement is quite inflexible and could be rewritten as follows: "The most common method for assessing vulnerability is the development of vulnerability curves that often ignores the characteristics of the buildings, especially when it comes to hazards involving a limited amount of buildings (e.g. debris flow), focusing mainly on the intensity of the process and the corresponding loss. Nevertheless, vulnerability curves for different types of buildings may be found in the literature for earthquake, wind and flood hazards".

Comment: Page2, line13: The definition of physical vulnerability is not in "conflict" with other, more general definitions of vulnerability. It is just one possible quantitative interpretation of it.

Reply: I will have to disagree with the referee at this point. The UNDRO (1984) definition is not a quantitative interpretation of the UNISDR (2009) definition. There is a big difference between them which lies on the fact that the one claims vulnerability to be loss (the result of a hazardous natural process-ex post) whereas the latter reflects the direct relationship of vulnerability to a pre-existing condition (ex-ante) of the element at risk. From these two types of definitions different methodologies derive that have different data needs (e.g. vulnerability curves need empirical data) and of course different results.

Comment: Page 18, line 14: The author seems to forget that physical vulnerability is basically the conditional probability of loss, given that a hazard of certain occurs. A building with high vulnerability will suffer little damage if it is subjected to low intensity hazard. Likewise, a building of low vulnerability may be totally destroyed during an extreme event.

Reply: The referee gives a definition of vulnerability which sounds more like a definition of risk. ("Risk is the probability of harmful consequences (...) resulting from

interactions between natural and human induced hazards and vulnerable conditions (UNDP-BCPR, 2004)). The term "probability" is relevant to the "fragility curves" (functions that describe the conditional probability that a damage state will be reached or exceeded for a given hazard intensity) which are not the focus of this paper. A distinction between vulnerability and fragility functions has not been made in the article, however, the author could do so if the editor thinks that it is necessary. The referee's statement following this definition has been also made by the author herself elsewhere in the text (page 18/ lines 18-23, page 20/ Line 13-15 and page 23/ lines 16-21).

Comment: Page 18, line 21: The indicator-based relative vulnerability index could be transformed to site-specific vulnerability curves if enough data exist for doing the transformation. When there is lack of data, the transformation could be based on expert engineering judgement. This of course involves some uncertainty, but the level of uncertainty is not necessarily greater than the variability observed when data are available (e.g. Figure 5 of the paper).

Reply: The referee here suggests that one could develop curves for different building types or buildings with different characteristics based on expert engineering judgement. This is a very interesting suggestion that is difficult to implement. First, as the referee also suggest the level of uncertainty will be very high but also the subjectivity of the method will limit the possibility of the curves to be used in other case studies. Second, this transformation as the referee also notes, would require a significant amount of data that for some hazard types are simply not available. Last but not least, the indicator based methodology considers a large amount of information regarding the buildings and not only the construction type or the number of floors. How would the combination of these characteristics be possible if we would attempt such a transformation?

Please also note the supplement to this comment:
http://www.nat-hazards-earth-syst-sci-discuss.net/nhess-2016-76/nhess-2016-76-

AC1-supplement.pdf

---

## Referee Comment (RC2) · Anonymous Referee #2 · 9 May 2016

Overall quality of the discussion paper ("general comments");

This study focuses on the application of an indicator-based methodology (IBM) for the assessment of physical vulnerability due to debris-flow hazards. The relatively "new" indicator-based method has been proposed, compared with the well-known vulnerability curves (highlighting weaknesses and strengths) and recommendations for its improvement outlined. It is really appreciable the effort made by the Author and the topic is of high value for the scientific community and within the scope of the Journal. Concerning the research, the Author is trying to compare and aggregate two methods that are funded on two different concepts. Vulnerability Curves (VCs) express damage in relation to hazard intensity. IBM mainly refers to the "intrinsic" susceptibility of

a building to suffer damage if affected by a specific type of event, independently from its magnitude. Generally speaking, it seems that IBM may be only useful to justify the final result, in terms of degree of loss. It has no predictive power: what may I expect if a 2.0 m debris flow event will happen in the future? On the contrary, this kind of information may be extracted from VCs and this is the type of information the authorities mainly require in their spatial planning and risk management activities. IBM is not a stand-alone procedure for vulnerability assessment and it needs information provided by VCs. Moreover, the use of indices undoubtedly increases the flexibility of the method but, at the same time, its subjectivity; and then, weighing indices makes the situation even worse by further raising the level of subjectivity. Probably, a sensitivity analysis has to be performed and results provided to check this important topic. As a general comment, a new methodology should provide clear approaches, rules of application, constraints, etc., in order to define a precise framework in vulnerability studies. On the contrary, in this research, the two methods have not been homogenized and integrated in a general framework but only "put one next to another". Probably, the original source of debate is that too many terms and too many concepts refer to the same thing. This is probably due to the fact that many components (or dimensions) composing vulnerability need to be investigated. Vulnerability is one of those terms that seems to defy consensus usage showing many different connotations, depending on the research orientation and perspective. The review of current vulnerability definitions demonstrates that, at least, two different perspectives exist: the former refers to an engineering and natural science point of view; the latter to a social science outlook. It all depends on the components (dimensions) of vulnerability each school of thought takes into account and privileges. A multi-disciplinary approach is a key-mode of actions in vulnerability studies and this allows each scholar to use his/her own terms and concepts, trying to homogenize them in a single framework. Sometimes less is more. In any case, from a practical point of view, it would be useful to know the real interest of local authorities to be aware of the intrinsic susceptibility of a building to suffer damage not considering the intensity of events. In my opinion, event intensity (and the return period, as well) is

a primary parameter also to find structural solutions for retrofitting buildings. And, in so doing, for decreasing their intrinsic susceptibility to suffer damage. The English sounds good (but I'm not a native speaker) although some expressions seem to be a little bit ambiguous and confusing; they have to be rephrased. The paper can be considered for publication after major revisions. This means that the Author should provide a clear framework by which each of the methods gives its real contributions in vulnerability studies. Moreover, if possible, self-citations should be a little bit reduced.

Individual scientific questions/issues ("specific comments");

The Author can find many other comments in the attached PDF file.

Please also note the supplement to this comment:
http://www.nat-hazards-earth-syst-sci-discuss.net/nhess-2016-76/nhess-2016-76-RC2-supplement.pdf

**Supplement:**

[revised manuscript text omitted]

---

## Author Comment (AC2) · 11 May 2016

**Response to Anonymous Referee #2**

First of all, I would like to thank the anonymous referee #2 for his remarks and his constructive comments. Following, you can find my response to the general and the specific comments.

**Reply to Interactive comment:**

The main requirement of the referee is to recognise that IBM is not a standalone method because it has no predictive power, however, it can improve and complement the existing VCs. This, according to the referee, should be shown in the paper in the form of a framework highlighting the contribution of each approach in vulnerability studies. My reply to each point of the referee in the general comments follows:

-Predictive power: I appreciate the comments of the referee and I totally agree with some statements, for example, "two methods that are funded on two different concepts" or "IBM mainly refers to the "intrinsic" susceptibility of a building to suffer damage". However, as far as the predictive power of the IBM is concerned, I have some objections. VCs are developed based on empirical data and are not transferable. IBM may be used where empirical data or curves are not available and has a predictive power in the sense that it may identify the relative vulnerable buildings in a qualitative way. It may not predict monetary loss (or degree of loss) but it may indicate the buildings that will experience loss based on their characteristics. There are numerous disadvantages of the IBM method and space for improvement but the fact that it can be applied where no empirical data are available is considered an advantage. Moreover, as it was also suggested by the referee, it may be used to improve existing VCs.

-IBM is not stand alone: IBM as a qualitative method of assessing relative physical vulnerability is standalone. However, it is true that the IBM may benefit from information coming from VC and this will be shown in the holistic framework (required by the referee) which will be included in the revised version.

-Flexibility vs. subjectivity: the issue of subjectivity has been discussed in the paper and solutions have been proposed (e.g. correlation of real damage data to building characteristics). A sensitivity analysis would be certainly worthwhile but would probably provide the material for the next paper following this one. I could refer to the sensitivity analysis, however, in the section of recommendations for improvement and future developments (page 24-25).

-"Too many concepts refer to the same thing": I agree with this statement and I recognise that a debate about vulnerability, different dimensions and definitions is missing mainly because it has been provided in previous papers by the author and colleagues. The revised version will definitely include more reference on this topic.

-Real interest of local authorities: The vulnerability curve presented in this paper was the product of very close collaboration of the local authorities in South Tyrol. The methodology for the development of the curve was based on the results of a stakeholder workshop. There was an effort to cover the needs of the end users and to also use their expert knowledge to develop the curve. There is definitely an interest from the side of the practitioners for the vulnerability curves and this is mainly because the curves provide a quantitative result.

Indicator based methodologies are not that attractive mainly because of the amount and detail of the data required.

-Importance of intensity: The fact that intensity is ignored has been recognised and discussed in the paper. The framework that will be included in the revised version will attempt to address this problem as well.

-The referee suggests that "the Author should provide a clear framework by which each of the methods gives its real contributions in vulnerability studies." The author appreciated the comment and will provide a framework which shows the contribution and the interactions o the methods in the revised version.

- self-citations should be a little bit reduced: this was also required by referee #1. I will try to reduce them.

**Reply to specific comments on the manuscript (supplement pdf)**

I would like to thank the anonymous referee for his comments regarding the spelling. I will make the recommended changes in the revised manuscript. As far as the other comments are concerned:

Abstract, line 18: The comment will be taken into consideration and the last sentence of the abstract will change as follows:

"*The comparison of the two methodological approaches and their results **is challenging since both approaches are dealing with vulnerability in a different way. However, it is still possible to** highlight their weaknesses and strengths **and to** show clearly that both methodologies are necessary for the assessment of physical vulnerability and emphasise the need for a "holistic methodological framework" for physical vulnerability assessment.*"

Page 2, line 3: I agree with the referee, however, there is a focus on debris flow (maybe I should stress this from the beginning of the paper) in the paper. Debris flow affects a limited amount of buildings and for this reason there is often not enough data to develop curves for each type of building. Nevertheless, even if we were able to do that for which characteristic of the building would we develop those curves? (for the building type? Age? Presence of openings? Surroundings?). In any case, a reference to the HAZUS curves is considered necessary (also from referee #1) and it will be included in the revised version.

Page 3, line 1: It is true that the sentence is general. I will remove the sentence in the revised version.

Page 3, line 3: I agree, I will remove "…reviewing methods for the development of vulnerability functions for tsunami"

Page 3, line 9: I will remove: "focusing on tsunamis"

Page 3, line 14: This is an interesting point that I will gladly include in the revised version.

Page 3, line 30: This is not about the specific study but about the use of indicators in general.

Page 4, line 34: this is exactly what I am saying "variety of building characteristics and surroundings" (the intensity is supposed to be the same)

Page 5, line 3: This issue is discussed later in the text (Page 24/lines 22-23, page 25/lines 5-7, Page 25, lines24-26)

Page 6, line 15: I will do so in the revised version

Page 7, line 4: The subjectivity issue is discussed later (Page 24/lines 22-23, page 25/lines 5-7, Page 25, lines24-26)

Page 7, line 4: I will add this reference at the beginning of the paper (page 2, line 10)

Page 8, Figure 2: No, there was no sensitivity analysis performed. I should probably clarify this in the text.

Page 11, Figure 5: The referee is right, however, for debris flow most curves to be found in the literature are lines and not buffered curves. A reference to this could be made in the next version.

Page 12, line 5: More information will be provided in the revised version.

Page 12, line 13: I agree with the referee. The comment will be considered in the revised version.

Page 15, line 17: more info will be added

Page 18, line 7: The referee is right, however, my statement is also right…

Page 18, line 17: The structures of the methods are indeed considerably different, however, they both claim to assess physical vulnerability and to provide a tool that can be used for risk analysis and ultimately for risk reduction. To which degree the two methods achieve that is one of the main aims of the paper.

Page 18, line 26: the fact that the two methodologies should complement and support each other is anyway one of the outcomes of the paper.

Page 18, line 27: This is true. Perhaps a good statement would be that: "although the intensity is not taken under consideration in IBM, information about intensity is hidden in some indicators (surroundings, building row etc.). However, such a statement has been already done at page 18/lines21-23.

Page 18, line 28: I do not agree 100%. The IBM does not have a predictive power when it comes to a specific scenario. However, the development of a vulnerability curve requires empirical data which are not always available. Transferring vulnerability curves from other places is possible only by similar housing design, materials and architecture. IBM could predict the relative vulnerability between buildings and indicate the ones that need reinforcement.

Page 18, line 31: This is exactly what the referee has already mentioned: although IBM is not a standalone procedure it could be used to support VC and on the other hand information derived by VC may support the IBM. For more detail see page 25, line 23 (Improved weighting of indicators).

Table 2: I appreciate the referee's point of view and I will consider it in the conclusions and discussion. I particularly like the suggestion that we do not need more methodologies but the improvement of existing ones.

Page 20, line 9: This is a point that the referee has made before in the text. It is true that "The efficiency of the integration of the methods still has to be proven". I will try to do so in the revised version. This is also a response to the point of the referee in the interactive comment "*the Author should provide a clear framework by which each of the methods gives its real contributions in vulnerability studies*".

Page 20, line 21: This is a comment which has been repeated often until now. It will be considered in the next version.

Table 3:

Measurable: I do not really understand the point of the referee. What does he or she mean with "event types"? hazard types (earthquakes, floods etc.)? It is probably my fault not to mention it earlier but we are talking about vulnerability to one hazard type at a time

Relevant: (C1) Not always, VCs require detailed empirical data which are not always directly accessible. Damage photos, for example, may give direct information about the relevance of indicators.

(C2) I can add this here although the comment has been also made elsewhere in the text

Policy-relevant: I agree with the comment and I will include it in the text.

Measure important: I agree with the comment and I will include it in the text.

Analytically and statistically sound: This statement Is probably quite strong. I could write instead: "Although the indicators may give an overview of the actual situation, the links…"

Understandable/easy to interpret: No, we do not want this but this is not the right place to say it. I keep the statement for the discussion or conclusion chapter.

Reproducible: At this point the indicators are tested against the criteria. The statement will be made elsewhere in the text.

Page 22, line 8: I agree and will add this.

Page 22, line 19: I address this point in my response to the interactive comment above

Page 22, line 26: I agree with comment and will add it to the text.

Page 23, line 5: Perhaps it is more correct to say "practitioners use vulnerability curves as a prediction tool rather than to acquire information about specific buildings in an area and for this reason they ignore their spatial component".

Page 23, line 8: I agree with the referee, the word "process" will be removed.

Page 23, line 21: No, because it includes additional information that the VC does not include.

Page 25, line 12: This is a statement that belongs, in my opinion, at the beginning of the paper (probably at page 2, line 10.)

Page 25, line 32: "a comparison of the two methods may shed light to their advantages and drawbacks and may also inform the practitioners on their available methodological choices". I believe that the paper is doing this up to a point. Both methods are used in the same area and even if their one to one comparison is impossible for reasons that the referee has also pointed out, they are both scrutinized and their advantages and disadvantages as well as recommendations for improvement are presented. The recommendation of the referee for a "*clear framework by which each of the methods gives its real contributions in vulnerability studies*" certainly adds a lot to this comparison.

Page 27, line 1: It is true. In my opinion VCs show the relationship between intensity and degree of loss based on empirical data. They do not refer to specific buildings and they absolutely ignore building characteristics. The practitioners may derive only the following piece of information. E.g. in case a debris flow of 2,5m impacts any given building the degree of loss will be 0,5. This does not include any information regarding the building so, it cannot guide any retrofitting for a specific building. The required information for something like that may be given only by the IBM.

Page 27, line 19: A general framework is a requirement of the referee and it will be attempted in the revised version of the paper.

Page 27, line 20: The fact that the curve becomes steeper after the intensity of 1,5m (where the lower level of the windows of the first floor usually is) reveals the importance of the existence of openings. The fact that there are points showing considerable degree of loss with low intensity is connected to the existence of a basement or basement windows that allowed the entrance of material within the basement and of course the occurrence of additional damage. Moreover, buildings that although have experienced high intensity, have not experience high degree of loss are usually buildings with higher initial value due to additional floors. All these observations show that number of floors, existence of basement and openings play a significant role in the amount of damage that a building will experience. These observations will be included in the revised version.

Page 27, line 23: yes, a holistic framework for physical vulnerability will be provided in the revised version.